# Shear Force Fiber Spinning: Process Parameter and Polymer Solution Property Considerations

**DOI:** 10.3390/polym11020294

**Published:** 2019-02-10

**Authors:** Arzan C. Dotivala, Kavya P. Puthuveetil, Christina Tang

**Affiliations:** Chemical and Life Science Engineering Department, Virginia Commonwealth University, Richmond, VA 23284-3028, USA; dotivalaac@vcu.edu (A.C.D.); kpputhuveetil@vcu.edu (K.P.P.)

**Keywords:** aligned fibers, electrospinning, entanglement, draw-down ratio

## Abstract

For application of polymer nanofibers (e.g., sensors, and scaffolds to study cell behavior) it is important to control the spatial orientation of the fibers. We compare the ability to align and pattern fibers using shear force fiber spinning, i.e. contacting a drop of polymer solution with a rotating collector to mechanically draw a fiber, with electrospinning onto a rotating drum. Using polystyrene as a model system, we observe that the fiber spacing using shear force fiber spinning was more uniform than electrospinning with the rotating drum with relative standard deviations of 18% and 39%, respectively. Importantly, the approaches are complementary as the fiber spacing achieved using electrospinning with the rotating drum was ~10 microns while fiber spacing achieved using shear force fiber spinning was ~250 microns. To expand to additional polymer systems, we use polymer entanglement and capillary number. Solution properties that favor large capillary numbers (>50) prevent droplet breakup to facilitate fiber formation. Draw-down ratio was useful for determining appropriate process conditions (flow rate, rotational speed of the collector) to achieve continuous formation of fibers. These rules of thumb for considering the polymer solution properties and process parameters are expected to expand use of this platform for creating hierarchical structures of multiple fiber layers for cell scaffolds and additional applications.

## 1. Introduction

Polymer nanofibers and microfibers have high specific surface area and porosity. Therefore, they are of potential interest in a range of applications including filtration, tissue engineering, smart clothing, reinforcement of composite materials, and electronics. [1,2,3,4,5]. Controlling the orientation of the fibers and hierarchical structures of multiple fiber layers is important for many applications including composite reinforcement as well as fabrication of functional fluidic, electronic, and photonic devices [6,7]. Such fiber structures are of considerable interest as cell scaffolds because fiber orientation and spacing affect cellular responses [8,9]. Controlling fiber orientation has been useful for vascular tissue [10], nerve regeneration [11], smooth muscle regeneration [12], etc. [13,14].

Aligned and patterned polymer nanofibers have been achieved via electrohydrodynamic fiber processing. In electrohydrodynamic processing (electrospinning), aligned nanofibers can be achieved by manipulating the electronic field. One approach has been to attenuate the bending instability during fiber processing. For example, applying a DC biased AC potential [15], employing dual-opposite spinnerets [16], or introducing counter electrodes [17] minimizes the bending instability and jet whipping during fiber processing so that aligned fibers are achieved. The collector also affects the electric field and can be manipulated to achieve aligned fibers. Gap and other patterned electrodes [7], concave surfaces [18], as well as rotating drums have been employed to align fibers [19,20].

Reducing the tip-to-collector distance from ~10 cm used in traditional electrospinning to ~1 mm for near-field electrospinning also reduces the bending instability. Utilizing this technique, deposition of fiber patterns is possible by manipulating the position of the collector during fiber processing [21,22]. This technique is particularly important for precise positioning of individual fibers for integrating fibers into devices such as optical sensors, light emitting diodes, and cell scaffolds [8,23].

Aligned and patterned polymer nanofibers have also been achieved using mechanical fiber spinning methods. The advantage of the mechanical methods is that they avoid the bending instability inherent to electrospinning processes. For example, centrifugal force can be used to elongate a drop of polymer solution into a fiber. Using this method, the rotational speed can be adjusted to align the resulting fibers [24]. Mechanical spinning of fibers has also been achieved by bringing a drop of polymer solution in contact with a rotating substrate. Touch-spinning [4] and spinneret-based tunable engineered parameters (STEP) [25] spinning setups have been investigated. Fiber alignment and patterning are achieved through controlled movement of the collector [4,25]. Analogous to near field electrospinning, precise, controlled placement of individual polymer fibers can be achieved by drawing a fiber from a pool of polymer solution using a stylus or direct-write continuous drawing. Fiber deposition and resulting patterning is controlled by positioning of the collector [26].

Using STEP, Nain and co-workers have systematically investigated the effect of process parameters on fiber diameter and fiber spacing [23,24]. The role of polymer entanglement has also been demonstrated [24]. Since polymer entanglement is affected by polymer molecular weight and concentration, empirical correlations of fiber size to molecular weight at constant concentration and fiber size to polymer concentration at a constant molecular weight for polystyrene/xylene systems have been explored [24]. Other systems, namely, polyurethane/dimethylformamide, poly(methyl methacryale)/chlorobenzene, poly(lactic-co-gycolic acid) (PLGA)/chloroform, fibrinogen/hexafluoroisopropanol/aqueous buffer, polyethylene oxide/ethanol/water have also been successfully shear force spun into fibers using STEP [23]. However, the number of systems that have been patterned with STEP has been limited relative to the number of systems that have been electrospun. The role of solvent properties, e.g. surface tension, of the polymer solution in shear force fiber spinning has not been fully described. 

We report a simple apparatus for shear force fiber spinning that avoids the use of highly advanced automated micropositioning systems and quantitatively assess the fiber spacing variability relative to electrospinning. The effect of solution properties on ability to form fibers is also discussed and compared with electrospinning and rotary jet spinning. Specifically, the use of capillary number and polymer entanglement concentration to guide selection of polymer solutions as well as a draw-down ratio to guide process parameter selection are examined.

## 2. Materials and Methods

### 2.1. Materials

Polystyrene (PS) (weight average *M*_W_ 350,000 g/mol), polyvinylpyrrolidone (PVP) (*M*_W_ 1,300,000 g/mol), polyvinyl alcohol (PVA) (*M*_W_ 205,000 g/mol, degree of hydrolysis 86.7%–88.7%), and polyethylene oxide (PEO) (*M*_W_ 2,000,000 g/mol) were received from Sigma Aldrich (St. Louis, MO, USA). Toluene (ACS reagent grade), acetone (ACS reagent grade), and ethanol (Molecular Biology Grade) were received from Fisher Scientific (St. Louis, MO, USA). All chemicals were used as received.

### 2.2. Solution Preparation

Various amounts (5–45 wt.%) of PS were dissolved in a mixture of toluene and acetone by stirring at room temperature overnight until macroscopically homogenous. Similarly, PVP was dissolved in ethanol, and PEO was dissolved in deionized water by stirring at room temperature overnight. Aqueous PVA solutions were prepared by stirring combinations of PVA and deionized water at 60 °C until macroscopically homogenous. All solutions were stored at 4 °C before further use.

### 2.3. Fiber Spinning

A shear force fiber spinning apparatus for 1D fiber patterning was built from a syringe pump (New Era Pump Systems Inc., Farmingdale, NY, USA), a lab jack, and two rotating motors (Pololu 50:1 Micro Gearmotor HPCB 12V and a NEMA-17 Stepper Motor) to rotate the collector and translate the jack vertically, respectively. The two motors were controlled using a using a Raspberry Pi control module. Collector speeds operate at a range 40 to 635 RPM, while the linear motion speed runs at a maximum speed of 1.48 mm/s. The collection substrate (1/16″ thick polyethylene sheet) is mounted in a 3D printed holder fixed to the DC motor and then secured atop the linear jack.

To spin fibers, polymer solution was pumped at ~0.5 mL/hr. The rotating collector was brought into contact with the drop of polymer solution at the tip of the needle tip (22G, 0.508 mm ID or 18G, 0.965 mm ID). After initial contact, the needle tip and collector were separated by 0.5″ to facilitate continuous fiber spinning. To pattern the 1D fibers, a layer of fibers was deposited as the collector completed a single pass by the needle (top to bottom or bottom to top). In some cases, the substrate was rotated and another layer was applied.

For comparison, fibers were electrospun using a rotating mandrel collector to align the fibers. Polymer solution was pumped (New Era Pump System, Inc., Farmingdale, NY, USA) through a 22G (0.508 mm ID) stainless steel needle (Jensen Global, Santa Barbara, CA, USA) at a constant rate while applying a constant voltage (Matsusada High Precision Inc., Shiga, Japan) to the needle. The mandrel (7/8″ diameter copper rod) was ground and rotated at 1700 rpm (using an electric motor). Typical process parameters were: tip-to-collector distance of 9 cm, applied voltage of 10–12 kV, and flow rate of 0.5 mL/h.

### 2.4. Characterization

#### 2.4.1. Solution Characterization

The zero-shear viscosity of polymer solutions was measured at 25 °C using a 40 mm parallel plate geometry and TA Instruments DHR-3 rheometer. The surface tension was measured using the pendant drop method [27,28] using a Rame-Hart Model 250 Goniometer with DROPimage software (Rame-hart instrument company, Succasunna, NJ, USA). The method for surface tension was calibrated with ethanol.

#### 2.4.2. Fiber Characterization

Fibers were imaged with optical microscopy (Nikon Eclipse LV100D, Nikon Instruments Inc., Melville, NY, USA). For higher resolution imaging, the fiber samples were coated with gold:palladium (60:40) and analyzed with scanning electron microscopy (SEM, Hitachi SU-70 FE-SEM, Hitachi, Chiyoda, Tokyo, Japan) with 5 kV accelerating voltage. The average fiber size and spacing (edge-to-edge) was determined from 20 to 30 measurements using ImageJ software (US NIH, Bethesda, MD, USA).

## 3. Results and Discussion

Traditional electrospinning and collecting the fibers on a rotating mandrel (Figure 1A) is a well-established, commonly used approach for achieving aligned nanofibers. To achieve a high degree of fiber alignment, high rotational speeds (~1000 rpm) of the mandrel are required. At 1700 rpm, aligned fibers were achieved and the fiber to fiber spacing was ~10 micron (Figure 1B). This result is comparable to previous reports achieved with the basic set-up [11,13] as well as variations of the basic set-up involving auxiliary electrodes [17], modified collectors [20], or translation of the needle [29]. Achieving larger fiber-to-fiber spacings, e.g. ~100 micron, which have been important for studying single cell behavior [30] have been reported using near-field electrospinning which involve both suppression of the whipping instability and translation of the collector [21,22].

Alternatively, we investigated shear force fiber spinning (which inherently avoids the bending instability) using a simplified setup with positional control of the rotating collector (Figure 1C) with the aim of avoiding highly advanced automated micropositioning systems. Rather, we utilized readily available, off-the shelf components to achieve programmable rotation and vertical translation of the collector. In this method, fiber formation is initiated by bringing the rotating collector in contact with the droplet of polymer solution. The shear force of this contact overcomes surface tension and the extruded polymer solution is continuously drawn. As the solvent evaporates, the fiber solidifies [25,31].

Fiber formation (Figure 1D) and deposition are well controlled by programmed movement of the collector. By controlling the movement of the collector, fiber spacings between 250 and 400 microns were achieved using polystyrene as a model system. Additionally, hierarchical fiber patterns (Figure 2) were achieved via layer-by-layer deposition similar to what has been achieved with setups using micropositioning systems [23]. Specifically, double-layer fiber arrays where the angle between the two layers were 90°, 45°, or 30° were achieved by varying the geometry of the collector. For example, orthogonal fibers were achieved with rectangular collectors. To produce the double layer fiber arrays, the first layer of fibers was deposited, the collector was rotated 90°, finally, a second, orthogonal layer of fibers was deposited. Triangular collectors can be used to layer fibers at 30° or 45° and rotating the collector between depositing layers. In Figure 2, the dotted lines represent the side of the collector that is placed in the holder for depositing the first layer (light blue) and second layer (dark blue). 

To examine the range of process parameter space (needle diameter, flow rate) that facilitated continuous fiber production, we varied the solvent system and mapped the process parameters. Specifically, we dissolved polystyrene in toluene or various mixture of toluene and acetone. For a given needle diameter (22G, ID 0.508 mm), we systematically varied the flow rate and rotational speed of the collector to determine the process parameters that resulted in continuous drawing of fibers (Figure 3A). Generally, systems with higher toluene content required a lower flow rate to continuously form fibers. This result suggests that the process parameters and solvent volatility should be considered together. Physically, these parameters correspond to the time scales of deformation due to elongational stresses and fiber solidification. 

To establish empirical rules of thumb for process parameter selection, we considered the draw-down ratio. The draw-down ratio (DDR) is defined as the ratio of the velocity of the collected fiber to the velocity at the spinneret face (Equation (1)). Based on previous reports [32,33,34], we approximate the DDR as:(1)$$DDR=\frac{v_{f}}{v_{i}}=\frac{\omega *2\pi w}{\dot{V}/\left(\pi r_{s}^{2}\right)}$$
where *w* is the width of the collector, *ω* is the rotational speed of the collector (RPM), *V* is the volumetric flow rate, and r_s_ is the radius of the spinneret. This calculation of DDR assumes no slippage of the fiber and the collector. Using 30 wt.% polystyrene in toluene or in toluene/acetone (8:2 *v*:*v*), we compare the draw-down ratios that result in continuous formation of fibers using two different needle sizes. Specifically, we compare a 22G (ID 0.508 mm) with a larger, 18G (ID 0.965 mm) needle. The process parameters for the two different needle sizes are compared in a box plot in Figure 3B. For both size needles, DDRs ~10^3^ were most likely to result in continuous fiber formation. Notably, using the larger needle size resulted in a smaller parameter space (flow rate and rotational speed of the collector) that formed fibers. Practically, smaller tip sizes may be beneficial for fiber formation. Overall, we found that continuous fiber formation can be achieved with draw-down ratios of ~10^3^ and is a convenient guide for adjusting process parameters for various polymer systems. 

Since 30 wt.% polystyrene in 7:3 *v*:*v* toluene to acetone using a 22G needle tip was observed to spin fibers for the widest range of draw-down ratios, we further explored the effect of process parameters on fiber spacing and fiber size. Deposition of the single, stable filament from the droplet provides unique control of fiber size and spacing. Fiber-to-fiber spacing could be readily tuned with the translational speed of the collector according to the relationship (Equation (2)):(2)$$l=\frac{1}{\omega }T$$
where *l* is the fiber-to-fiber spacing (mm), *ω* is the rotational speed (revolutions/second), and *T* is the translational speed of the collector (mm/second) [31]. At a constant rotational speed, we observed a linear relationship between the fiber spacing and the translational speed of the collector as expected (Figure 4). The slope of the line of 0.39 s was comparable to the expected value of 0.35 s related to the rotational speed used (170 RPM).

We used image analysis to quantify the uniformity of the fiber spacing. We compare the relative standard deviation of the fiber spacing obtained from either shear force fiber spinning or electrospinning with the mandrel. With electrospinning, the average spacing was 11 microns with a relative standard deviation of 39% (Figure 5). Using shear force fiber spacing, the fiber spacing was 230 microns with a relative standard deviation of 18% (at a translational speed of 0.6 mm/s) (Figure 5). The maximum fiber spacing achieved by increasing the translational velocity of the collector to 1.1 mm/s was 510 microns (24% relative standard deviation). While the variation in spacing is larger than the 7% average fiber spacing variation reported using a micropositioning setup [8], this result demonstrates the expected trade-off between set-up simplicity and spacing precision. Impressively, the fiber spacing without using a micropositioning setup is more uniform than electrospinning as indicated by the ~2-fold reduction in relative standard deviation. It is also important to note that this method for shear force fiber spinning is complementary to electrospinning techniques as the fiber spacings that are achieved are an order of magnitude higher (~100 micron compared to 10 micron). The larger fiber spacings have been of interest for studying single cell behavior [30].

We also investigated the effect of process parameters on the resulting microstructure of the deposited fibers. Specifically, we examined the fiber size in addition to the fiber to fiber spacing. We observe an approximately 5-fold decrease in fiber diameter when increasing rotational speed of the collector from 450–620 rpm (Figure 6). This trend has been previously observed and attributed to higher forces on the polymer chains at higher rotational speeds leading to larger deformation of the polymer chains and ultimately smaller diameter fibers [17,24]. We note that varying the rotational speed of the collector was more effective at tuning fiber diameter than other process parameters (e.g., needle tip size).

Next, we were interested in expanding to additional polymer systems Since sufficient concentration for polymer entanglement is required for fiber formation using shear force fiber spinning [23,25] and polymer entanglement is also important for electrospinning uniform fibers [28,30], we focus on systems that have been commonly electrospun to investigate the effect of solution properties on shear force fiber spinning.

To compare with electrospinnable systems, we estimated the entanglement concentration for polystyrene in toluene/acetone (7:3 *v*:*v*) mixtures by analyzing the specific viscosity as a function of polymer concentration. For neutral polymers in good solvents, the scaling of the specific viscosity with polymer concentration changes with the onset of polymer entanglement [35,36,37]. For the model system used here, we estimate the entanglement concentration (C_e_) to be 8 wt.% based on the change in scaling from *η*_sp_ ~ c^2.6^ to *η*_sp_ ~ c^4.4^. We observe the ability to spin uniform fibers at 28 wt.% which is ~3*C_e_. Although this result is slightly higher than 2–2.5*C_e_ previously reported for electrospinning neutral polymers, it suggests that entangled, electrospinnable systems are a reasonable starting point for shear force fiber spinning. 

Therefore, we next explored fiber formation using additional polymer/solvent systems based on their ability to be electrospun. Specifically, we examine PEO/water, PVA/water, and PVP/ethanol system, which all have been reported to be electrospinnable [38,39]. PEO/water and PVP ethanol fibers were achieved using shear force fiber spinning. However, fibers were not obtained from 9.5 wt.% PVA/water at any process parameters (DDR ~ 60–1500) despite entanglement (C_e_ ~ 2.5 wt.%) and ability to achieve fibers via electrospinning [40]. Higher concentrations were not soluble. Since PEO and PVA had comparable concentrations in water (i.e. comparable solvent volatility), the draw ratios attempted were expected to be appropriate to achieve fibers from both systems. However, PVA fibers could not be achieved. This result indicates that entanglement is not necessarily sufficient for fiber formation. 

Therefore, we next considered additional solution properties that may affect fiber formation. This method of fiber spinning is initiated as the shear force from contact with the rotating collector overcomes surface tension. We compared the surface tension of the various polymer solutions using the pendant drop method (Table 1). The aqueous PVA solution that did not form fibers had the highest observed surface tension; it was approximately 2–3 fold higher than the other polymer solutions that successfully formed fibers. Thus, we surmise the shear force required for overcoming surface tension induces droplet breakup rather than drawing into fibers.

The capillary number (Ca) has been used previously to estimate jet break-up during fiber formation from polymer solutions. It characterizes the ratio of the viscous force to the surface tension force and is defined as
(3)$$Ca=\frac{\eta U}{\gamma }$$
where *η* is the dynamic viscosity, *γ* is the surface tension and U is the estimated jet speed [41]. The ratio of viscosity to surface tension for PVA is ~10-fold lower than the other polymer solutions. For PVA, Ca ~5 and for the model polystyrene system Ca ~50 (Table 1). These results are comparable to previous reports for Rotary Jet-Spinning. In that approach, the polymer jet is initiated using centrifugal forces and continuous fibers are achieved with Ca ~10–50. The lower capillary number results in shorter jet lengths and earlier break-up into droplets, which prevents fiber spinning [41].

This result indicates a similar underlying mechanism of fiber formation to rotary jet spinning. While rotary jet spinning has the capacity for higher throughput, shear force fiber spinning enables control over fiber to fiber spacing that has not been demonstrated with rotary jet spinning. 

Overall, electrospinnability and entanglement are a good starting point for identifying potential polymer solutions that will form fibers using shear force fiber spinning. The surface tension must also be carefully considered as high surface tension can prevent fiber spinning. Therefore, polymer entanglement and capillary number can serve as a guide for expanding the range of materials that can be patterned using shear force fiber spinning. For determining appropriate process parameters, the draw-down ratio is an important consideration.

## 4. Conclusions

To align and pattern polymer fibers, we compare shear force fiber spinning with electrospinning with a rotating drum collector. The fiber-to-fiber spacing using shear force fiber spinning is larger (~100 micron) than commonly used electrospinning techniques (<10 micron) and the fiber space variation is lower than electrospinning. To expand the range of materials that can patterned, we use polymer entanglement and capillary number as a guide for formulation of the polymer solution. The draw-dawn ratio guides process parameter selection. These rules of thumb considering the polymer solution properties and process parameters are expected to expand the material selection and potential applications of this platform for creating hierarchical structures of multiple fiber layers.

## Figures and Tables

**Figure 1 polymers-11-00294-f001:**
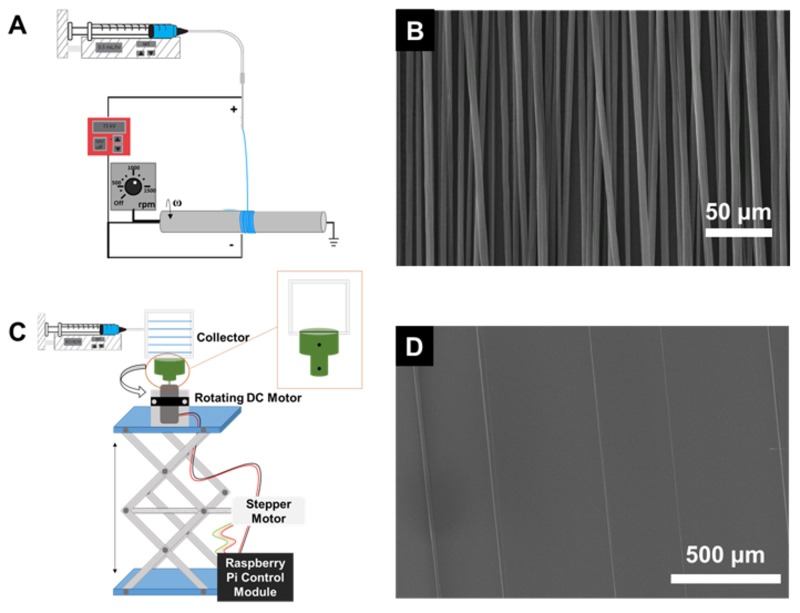
(**A**) Schematic for creating aligned fibers by electrospinning with a rotating mandrel. (**B**) SEM image of the aligned electrospun fibers. (**C**) Schematic for creating aligned fibers by shear force fiber spinning. (**D**) SEM image of the aligned fibers from shear force fiber spinning.

**Figure 2 polymers-11-00294-f002:**
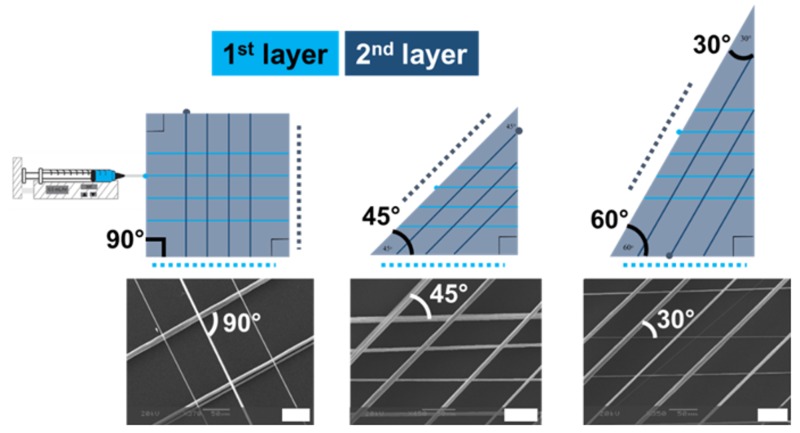
Representative hierarchical structures of multiple fiber layers with 50 micron scale bar. The angle between the fibers was dictated by collector geometry. Orthogonal fibers were achieved with rectangular collectors. The first layer of fibers was deposited, the collector was rotated 90°, finally, a second, orthogonal layer of fibers was deposited. Triangular collectors were used to layer fibers at 30° or 45°. The dotted lines represent the side of the collector that is placed in the holder for depositing the first layer (light blue) and second layer (dark blue).

**Figure 3 polymers-11-00294-f003:**
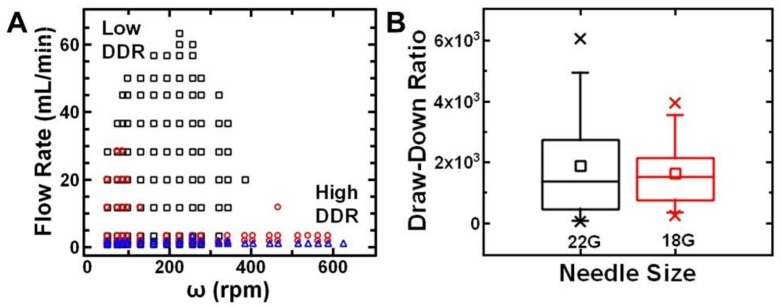
(**A**) Map of spinnability of 30 wt.% polystyrene solutions at varying flow rates and collector speeds for 7:3 *v*:*v* toluene to acetone (black squares), 8:2 *v*:*v* toluene to acetone (red circles), and toluene (blue triangle) (**B**) Box plot of draw-down ratio ranges for 30 wt.% polystyrene in 8:2 *v*:*v* toluene to acetone. Draw-down ratios of ~2000 are appropriate for continuous processing of polystyrene fibers.

**Figure 4 polymers-11-00294-f004:**
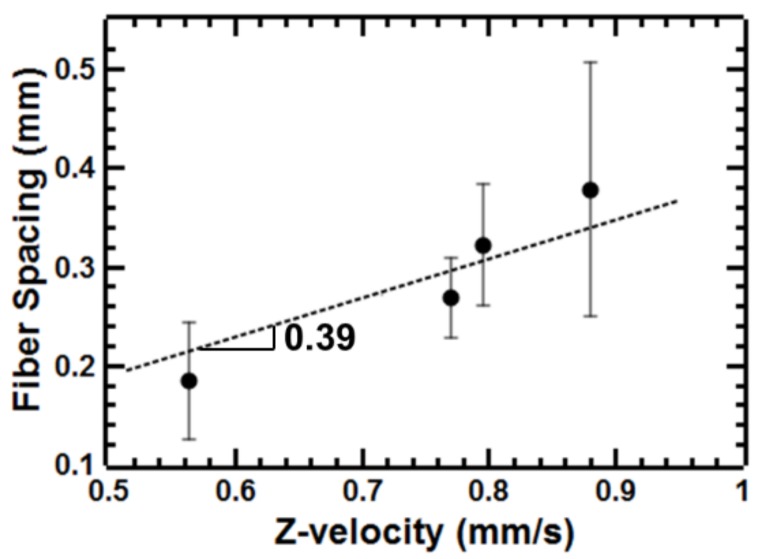
Effect of translation velocity of the collector on fiber to fiber spacing using 30 wt.% polystyrene (PS) in 7:3 toluene:acetone and a 22G needle tip as a model system.

**Figure 5 polymers-11-00294-f005:**
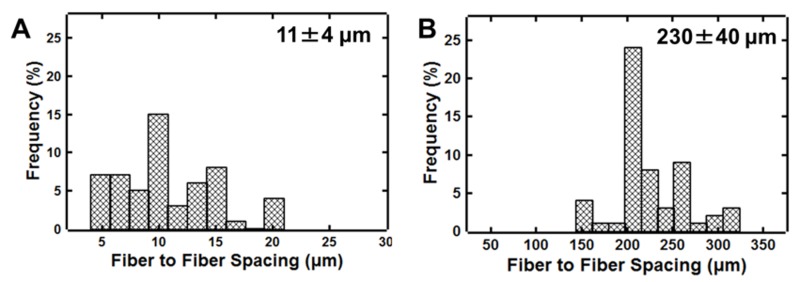
Fiber to fiber spacing distributions achieved by (**A**) electrospinning with a mandrel and (**B**) shear force fiber spinning using 30 wt.% polystyrene in 7:3 *v*:*v* toluene:acetone. The relative standard deviation of fiber using shear force fiber spacing is 2-fold lower than electrospinning with a mandrel.

**Figure 6 polymers-11-00294-f006:**
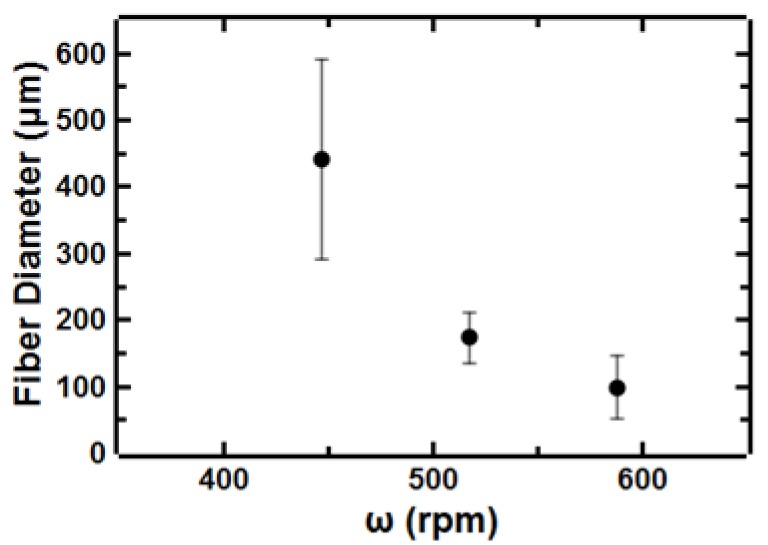
Effect of rotational speed of the collector on fiber diameter 30 wt.% polystyrene in 7:3 toluene:acetone and a 22G needle tip as a model system.

**Table 1 polymers-11-00294-t001:** Polymer solution properties and capillary numbers. PEO: polyethylene oxide; PVP: polyvinylpyrrolidone; PVA: polyvinyl alcohol.

Polymer Solution	*η* (Pa∙s)	*γ* (mN/m)	Ca	Fibers
Polystyrene/Toluene/Acetone	4.5 ± 0.3	13 ± 2	60	Yes
PEO/water	140 ± 30	22 ± 2	1000	Yes
PVP/ethanol	2.9 ± 0.3	27 ± 5	20	Yes
PVA/water	1.5 ± 0.1	43 ± 9	6	No

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
