# Peer review of "Shear Force Fiber Spinning: Process Parameter and Polymer Solution Property Considerations"

_polymers, 2019, doi:10.3390/polym11020294_

Round 1

Reviewer 1 Report

Dear Editor,      

The manuscript presents the shear-forced fiber spinning to align and pattern the polymeric nanofiber and its parameters for well-spun nanofiber. The authors prepared the aligned polymeric nanofibers using both electrospinning and shear-forced fiber spinning and then, compared the characteristics of them. In addition, the authors analyzed the processing parameters to spin polymeric nanofiber, and then also compared the polymer solution properties for spinning the aligned nanofiber. Overall, the experiments have been well-designed and the authors have performed systematic analysis for conclusion. However, I would recommend several alternations before final publication.

Q1. In overall manuscript, there are some typos. i.e. ‘Electrospinning with the, the average’ in line 198. The authors must revise the manuscript.

Q2. The authors compared the gap distance between fibers which were prepared by electrospinning and shear-forced fiber spinning, and represented the shear-force spinning has more advantage for the aligned nanofiber than the electrospinning from line 127 to 143. But it does not make sense for us. In general, the electrospinning process such as electrohydrodynamic-jetting can control large and small gap distance of nanofibers using movement of collector. In addition, I don’t understand why the authors compared the electrospinning and shear-forced spinning system without same movement system. (the electrospinning system does not include movement system of the collector.) To correctly compare these system, the electrospinning system must accompany the step movement system.

Q3. The figure 1 B and D are not suitable to compare the electrospinning and the shear-forced fiber spinning because the images were taken by different methods. Therefore, I would recommend to add the optical microscope image of electrospinning and SEM image of shear-forced fiber spinning in same resolution.

Q4. The discussion for the figure 2 is not enough. The authors must describe more discussion about figure 2.

Reviewer 2 Report

The manuscript “Shear Force Fiber Spinning: Process Parameter and Polymer Solution Property Considerations” presents interesting results that contributes to the processing of polymer fibers arrays. However, there are some minor points that need to be addressed

1.       The authors used the electrospinning as control, and several polymer systems to study them under the shear force fiber spinning process. Did the authors achieve fibers of PVA system under electrospinning process? Can the authors add the hydrolyzed grade of PVA besides MW?  

2.       The authors claim that using surfactants reduce surface tension and might facilitate fiber formation, however, they did not present any experiment or reference that support the statement.

3. Typos

Page 5 line171 the format of V

Page 5 line172 rs subscript

Page 5 line177-178 revise this phrase: “Notably, the larger needle size reduced the parameter space (flow rate and rotational speed of the collector that formed fibers”

Page 8: Revise the AKNOLEDGMENTS

Reviewer 3 Report

Authors in this manuscript introduced a novel spinning apparatus for shear force spinning, and investigated the effects of process parameters and polymer solution properties on the spinning process and fiber formation. In general, the manuscript is well-written; the results and conclusion are supported by the experimental. This reviewer suggests an acceptance for this manuscript.

Author Response

We thank the reviewer for their comments and are pleased that they consider the manuscript well-written.

Round 2

Reviewer 1 Report

The authors presented the shear force spinning in this manuscript, and investigated the effects of experimental parameters to spin well-designed nanofiber structure. In general, this manuscript is well-revised, and the results and conclusions are well-discussed. Thus, this reviewer suggests an acceptance for this manuscript.